# Gene profiles of peripheral white blood cells as potential predictors of pregnancy in embryo-recipient heifers

Mariam Raliou[1,2]*, Marie Margarete Meyerholz-Wohllebe[3,4], Doulaye Dembélé[5], Kirsten Mense[3,6], Maike Heppelmann[3], Christophe Richard[1,2], Pascale Chavatte-Palmer[1,2], Isabelle Dieuzy-Labaye[7¤], David Smith[8], Peter Zieger[9], Hans-Joachim Schuberth[10], Marion Schmicke[3], Iain Martin Sheldon[11], Olivier Sandra[1,2]*

**1** Université Paris-Saclay, UVSQ, INRAE, BREED, Jouy-en-Josas, France, **2** Ecole Nationale Vétérinaire d'Alfort, BREED, Maisons-Alfort, France, **3** University of Veterinary Medicine Hannover, Foundation, Clinic for Cattle, Endocrinology Laboratory, Hannover, Germany, **4** Clinic for Ruminants with Ambulatory and Herd Health Services, Centre for Clinical Veterinary Medicine, Ludwig-Maximilians-University Munich, Oberschleißheim, Germany, **5** Institut de Génétique et de Biologie Moléculaire et Cellulaire, CNRS UMR - Inserm U 964 - Université de Strasbourg, Illkirch, France, **6** Masterrind GmbH., Verden, Germany, **7** Zoetis, Zaventem, Belgium, **8** Independent Microbiologist, Edinburgh, United Kingdom, **9** Förster-Technik GmbH., Engen, Germany, **10** Institute of Immunology, University of Veterinary Medicine Foundation Hannover, Hannover, Germany, **11** Swansea University Medical School, Swansea University, Swansea, United Kingdom

¤ Current address: OIE Organisation Mondiale de la Santé Animale, 12, rue de Prony 75017 Paris, France
* mariam.raliou@inrae.fr (MR); olivier.sandra@inrae.fr (OS)

## Abstract

The bovine endometrium undergoes cellular, molecular, and functional changes to support embryo survival and development to term. These changes involve a finely coordinated series of events at both local and systemic levels. We postulated that circulating white blood cells (WBCs) could provide valuable biomarkers for predicting pregnancy success in heifers undergoing embryo transfer, when, sampled during both a preconception cycle and a conception cycle, before embryo transfer takes place. WBCs were isolated using PAXgene Blood RNA tubes collected from Holstein-Friesian heifers on Days 7 and 14 of a preconception estrous cycle (PCD7 and PCD14) and, after a rest cycle, on Day 7 of the subsequent estrous cycle (1ETD7) just before embryo transfer. Circulating progesterone and estrogens were assayed and pregnancy was confirmed by either uterine flushing and conceptus collection on Day 18 post-estrus, or the delivery of a healthy calf. Using a custom bovine gene expression microarray representing 19,479 unique transcript, comparison of transcriptomes between heifers classified as non-pregnant or pregnant revealed 1,240, 896 and 1,023 differentially expressed genes (DEGs) at PCD7, PCD14 and 1ETD7 respectively. Our bioinformatics analyses revealed that pregnancy failure after embryo transfer was associated with upstream regulators, biological functions, canonical pathways and gene networks related to inflammation, immunity, apoptosis and cell death regulation, cell proliferation, membrane compounds, lipid metabolism,

**Data availability statement:** All transcriptomic profile data files are available in the GEO database (accession number: GSE185325).

**Funding:** The study was funded by the ERA-NET EMIDA (Coordination of European Research on Emerging and Major Infectious Diseases of Livestock), 2010. Iain Martin Sheldon was supported by the UK Biotechnology and Biological Sciences Research Council Grant BB/1017240/1.

**Competing interests:** The authors have declared that no competing interests exist.

oxygen transport and ions transport. The heifers classified as non-pregnant showed significant increased transcripts levels of *PTGR1* at the three time points (PCD7, PCD14 and 1ETD7), *AIF1* at PCD14, *FNDC3B*, *IL15* and *SERPINE1* at 1ETD7. Our findings highlight the potential of peripheral WBCs as a non-invasive source of biomarkers for predicting pregnancy outcomes, offering promising insights for improving pregnancy success when reproductive biotechnologies are used in mammalian females.

## Introduction

In mammals, successful implantation requires finely coordinated paracrine and endocrine signaling between the embryo and the receptive uterus, leading to significant physiological and molecular changes in the maternal endometrium, including immune system modulation [1–5]. The endometrium plays an essential role in implantation and the establishment of pregnancy, with key genes identified as markers of endometrial receptivity in cattle [6–14]. In addition to these molecular markers, circulating immune cells are recruited to the uterus, where they facilitate interactions between the maternal environment and the early conceptus by preventing infections, eliminating apoptotic cells, promoting immunotolerance, and supporting conceptus development [15,16]. In response to the presence of the embryo, the regulation of immune cells such as monocytes, macrophages, and dendritic cells triggers the upregulation of cytokines like IL-12B and IL-15, and a downregulation of IL-18 in the bovine endometrium [17]. Trophectoderm cells of the conceptus secrete interferon-tau (IFNT), which is the key signal for pregnancy recognition in ruminants. IFNT maintains progesterone production by inhibiting luteolysis and modulates the maternal immune response [18].

During the peri-implantation period, IFNT-stimulated genes are significantly elevated in the plasma of pregnant cows, while pro-inflammatory cytokines (such as IL-8 and TNF-α) and adhesion molecules (like CD62L and CD11b) show higher expression in non-pregnant cows [2]. In women, studies suggest that immune cells, in concert with the endocrine system, facilitate early embryo-maternal interactions [19]. Interestingly, lymphocyte subpopulations such as CD56+ and CD16+CD56+ cells are more abundant in the peripheral blood of women who experience implantation failure compared to those with successful implantation [20]. These findings underscore the critical role of circulating immune cells in preparing the maternal environment for embryo survival and pregnancy maintenance.

The endometrium acts as a biological sensor and regulator of pregnancy success, influencing embryo development and adapting to long-term changes [21,22]. Considering the close interactions between the endometrium and circulating immune cells [23], systemic phenotyping of immune cell profiles could help predict a female's ability to become and remain pregnant [24]. For example, transcriptome profiles of peripheral blood mononuclear cells (PBMCs) collected before artificial insemination correlate with fertility in beef heifers [3]. Intrauterine administration of autologous

PBMCs during the estrous cycle has been shown to improve pregnancy rates following bovine embryo transfer [25]. Similarly, the intrauterine administration of PBMCs from non-pregnant mice before embryo implantation enhanced endometrial receptivity and implantation in mice with implantation dysfunction [26]. Furthermore, PBMCs administration in women with repeated in vitro fertilization (IVF) failure has been reported to improve embryo implantation [27] and in women with repeated implantation failure (RIF), PBMCs decrease the expression of estrogen receptor α (ERα) and progesterone receptor (PR), enhancing endometrial receptivity during the implantation window [28].

During gestation, the endometrium undergoes changes in immune cell populations, with increases in $CD14^+$ and $CD172a^-CD11c^+$ cells and elevated expression of cytokine genes such as IL-12B and IL-15, while IL-8 transcript levels decrease [17]. Increased levels of inflammatory cytokines such as TNF-α and IFN-γ in PBMCs are associated with infertility [29]. Among the cytokines and immune cell-attracting chemokines, IL-15 plays a key role in recruiting natural killer cells to the endometrium, where they differentiate into uterine natural killer cells, aiding implantation in both mice and humans [30]. These findings demonstrate the importance of cytokines and immune cells in mediating communication between the uterus and the peripheral immune system.

The corpus luteum plays a crucial role in regulating the immune system to create a favorable maternal environment for pregnancy [31]. Steroid hormones regulate the production of growth factors, cytokines, lipid mediators, and transcription factors in the endometrium [32]. In the bovine endometrium, progesterone and estrogen are key regulators of gene expression related to immunity, angiogenesis, tissue homeostasis, metabolism, and transport processes [12,33]. A recent study found that variations in the expression of bovine endometrial genes related to extracellular matrix interactions, histotroph composition, prostaglandin synthesis, TGF-β signaling, inflammation, and leukocyte activation on Day 7 of the estrous cycle are crucial for conception [7]. Similarly, changes in immune cells and cytokine gene expression in the human endometrium are essential for successful conception [34]. These findings collectively highlight the vital role of immune cells in preparing the uterus for pregnancy.

Circulating immune cells are influenced by the physiological state of the uterus and may serve as indicators of a female's ability to support pregnancy to term. In cattle, early embryonic mortality, often occurring between 8 and 16 days post-insemination, reduces production efficiency [35]. Early embryonic loss during the pre-implantation period is often linked to a compromised endometrial environment or embryonic incompetence [36]. In the present study, we hypothesize that gene expression in peripheral white blood cells (WBCs) during the estrous cycle may predict a female's capacity to initiate and sustain pregnancy after embryo transfer.

The mRNA profiles were analyzed from WBCs of heifers that did not achieve pregnancy after two successive embryo transfers (non-pregnant heifers) and those that became pregnant after the first transfer (pregnant heifers). Samples were collected on Days 7 and 14 post-estrus during the preconception cycle, as well as on Day 7 post-estrus during the conception cycle.

## Materials and methods

### Ethics statement

The study was conducted at the clinic for Cattle, University of Veterinary Medicine, Foundation, Hanover with the authorization of the German legislation on animal welfare (Lower Saxony Federal State Office for Consumer Protection and Food Safety, AZ 33.14-42502-04-12/0744). All procedures involving animals were carried out in accordance with German legislation on animal welfare ("Tierschutzgesetz", https://www.gesetze-im-internet.de/tierschg/BJNR012770972.html).

### Animal experiments

Thirty-two Holstein-Friesian heifers (Masterrind, Germany) were purchased from a single farm in Denmark. Two heifers were excluded from the experiment due to abnormalities of the reproductive tract. The thirty heifers enrolled in this study had a mean age of $14.1\pm1.6$ months and a mean body weight of $337.8\pm23.7\,kg$ at the start of the experiment. Before the

initiation of the experiment, the animals underwent a two-week adaptation period. The conditions for the maintenance and monitoring of the animals were previously described by Meyerholz et al. [37].

The experimental protocol is summarized in Fig 1. Animals underwent serial estrus synchronization and two embryo transfers (ET). To induce synchronized ovulation an intramuscular injection of 2 ml prostaglandin $F_{2\alpha}$ (Estrumate; Intervet, Germany), at a concentration of 250 µg/mL of cloprostenol, was administered, followed 48 hours later by a second injection of 2 ml gonadotropin-releasing hormone (Receptal; Intervet, Germany), at a concentration of 0.004 mg/ mL of buserelin. The heifers were repeatedly monitored for estrus detection, and clinical signs were recorded by a qualified veterinarian. Transrectal ultrasound monitored ovulation twice daily (LOGIQ Book XP, linear transducer probe 10 MHz; General Electric Medical Systems, China). If ovulation was not detected, the synchronization protocol was repeated.

The heifers were fed according to standard farming practices, and body temperature measurements were taken using rectal thermometers to assess potential abnormalities related to the hormonal treatment.

After the first preconception synchronization cycle, animals underwent two consecutive synchronization cycles with embryo transfer (ET). On Day 7 post-estrus, ET was performed in suitable recipients with a corpus luteum > 1.5 cm and serum progesterone > 1.0 ng/ml (Immulite System, Siemens Healthcare Diagnostics, USA; intra-assay CV ranged from 6.3 to 16%, inter-assay CV ranged from 5.8 to 16%, and analytical sensitivity was 0.09 ng/ml), as previously described [37,38]. Specifically, on Day 7 post-estrus, synchronized recipient heifers for embryo transfer were administered 3–4 mL of epidural lidocaine (20 mg/mL concentration, Xylovet®, CEVA, France). An embryo transfer gun was used to deposit the embryo into the uterine horn on the side of the corpus luteum.

All transferred embryos were male, produced, graded 2+ or 2-, and sexed by Masterrind GmbH (Verden, Germany). Embryos were frozen in glycerol and stored in liquid nitrogen, then thawed under sterile conditions using the Emcare Embryo Thawing System (ICPbio Reproduction, USA). One embryo was transferred trans-cervically using an embryo transfer instrument (Transfer Stylet 19240/1000, Transfit Lateral 19240/1005, Sanitary Sheaths 19271/0080; Minitüb, Germany) and deposited into the uterine horn on the side of the corpus luteum.

## Pregnancy detection

Pregnancy success was determined on Day 18 post-estrus through circulating progesterone (P4) measurement, uterine flushing and conceptus collection, as described previously [38]. In detail, donor heifers received a 3–4 mL epidural injection of lidocaine (20 mg/mL, Xylovet®, CEVA, France) to reduce discomfort. A cervical dilator was gently inserted and maintained for a few minutes before introducing a Foley catheter for uterine flushing. The flushing medium (Euroflush®, IMV Technologies) was pre-warmed to 38 °C and kept at a constant 30 °C in a thermostated box throughout the collection process. Ovaries were palpated rectally to confirm the presence of corpora lutea. All animals were handled calmly and monitored post-procedure, with no signs of pain or complications observed.

Animals were considered pregnant if circulating progesterone (P4) ≥ 1 ng/ml and an elongated blastocyst with an embryonic disc was present. Non-pregnant females had P4 < 1 ng/ml and no conceptus flushed. Following the flushing, non-pregnant animals received a 2 ml injection of prostaglandin $F_{2\alpha}$ (250 µg/mL of cloprostenol) to induce a physiological estrous cycle and were rested for 1 month before a second synchronization and embryo transfer. Four pregnant females from the first ET were allowed to carry to term and were classified as pregnant (n = 4). Heifers failing to conceive after two consecutive embryo transfers were classified as non-pregnant (n = 9). Animals with discordant results (P4 ≥ 1 ng/ml but no conceptus collected; n = 1 in the first estrous cycle; n = 3 in the second) and with embryonic mortality (n = 2) were excluded from further analysis.

## Progesterone and estrogens beta assays

Serum progesterone and estrogens concentration were determined in heifers categorized as non-pregnant (n = 9) and pregnant (n = 4), as previously described [37,39] and as shown in Table 1. The serum levels of progesterone were

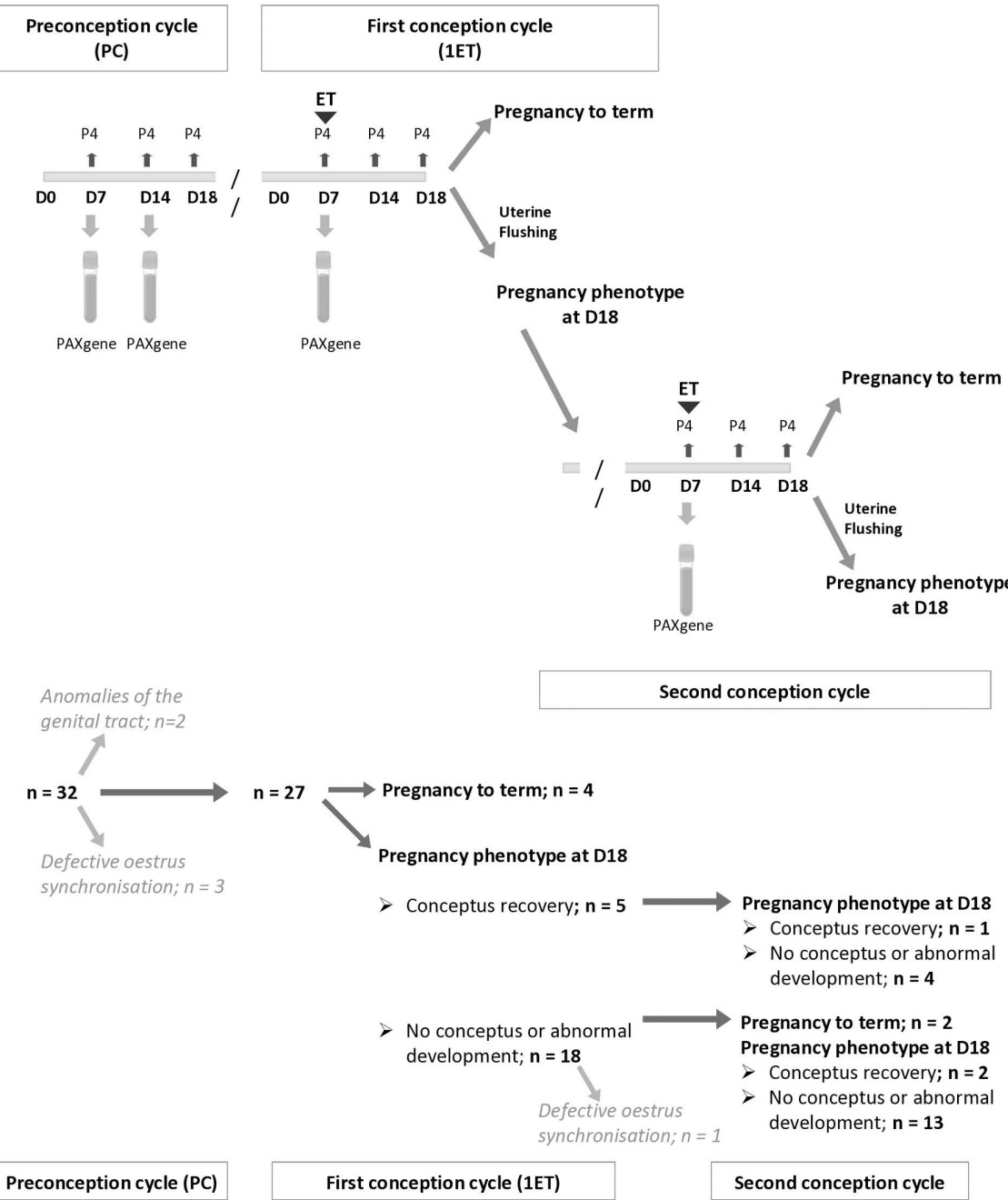

**Fig 1. Overview of the experimental procedures and heifer phenotypes.** (**Upper panel**): Estrus synchronization without embryo transfer defined the preconception cycle (PC). After completing the PC, females were left undisturbed for one month before undergoing a second estrus synchronization, this time combined with the transfer of a single male embryo on Day 7 post-estrus (D7). Heifers in this conception cycle (1ET) were either carried to term pregnancy or subjected to uterine flushing on Day 18 post-estrus (D18) to recover the conceptus. Following uterine flushing, the females were again left for one month before entering a third estrus synchronization, which included the transfer of another embryo on D7. Pregnancy status in this second conception cycle was assessed on D18 via uterine flushing. For each cycle, blood samples were collected on Days 7 (D7), 14 (D14), and 18 (D18) to measure circulating progesterone (P4) and estrogens levels. For microarray analyses, peripheral blood was collected using PAXgene blood RNA tubes on D7 and D14 of the PC, as well as on D7 of the 1ET cycle (1ETD7). (**Lower panel**): For each cycle, numbers of heifers and their phenotype is indicated.

Table 1. Details of Holstein-Friesian heifers used for serum progesterone and estrogens assays, microarray hybridization and RT-qPCR validation.

| Animal ID (Heifer: H) | Physiological status of animals after two embryo transfer | Plasma progesterone and estrogens assay | | | Microarray hybridation/analyses | | | RT-PCR validation | | |
|---|---|---|---|---|---|---|---|---|---|---|
| | | Preconception cycle/days post-estrus (PCD) | | First embryo transfer/days post-estrus (1ETD) | Preconception cycle/days post-estrus (PCD) | | First embryo transfer/days post-estrus (1ETD) | Preconception cycle/days post-estrus (PCD) | | First embryo transfer/days post-estrus |
| | | PCD7 | PCD14 | 1ETD7 | PCD7 | PCD14 | 1ETD7 | PCD7 | PCD14 | 1ETD7 |
| H2 | Non-pregnant (n = 9) | X | X | X | | | X | X | X | X |
| H12 | | X | X | X | | | X | X | X | X |
| H13 | | X | X | X | X | X | X | X | X | X |
| H14 | | X | X | X | X | X | X | X | X | X |
| H15 | | X | X | X | X | X | X | X | X | X |
| H18 | | X | X | X | X | X | | X | X | |
| H21 | | X | X | X | X | X | | X | X | |
| H23 | | X | X | X | X | X | | X | X | |
| H32 | | X | X | X | X | X | | X | X | |
| | | | | | | | | | | |
| H1 | Pregnant (n = 4) | X | X | X | X | X | X | X | X | X |
| H11 | | X | X | X | X | X | X | X | X | X |
| H19 | | X | X | X | X | X | X | X | X | X |
| H22 | | X | X | X | X | X | | X | X | X |

measured with a solid-phase radioimmunoassay (TKPG1, Coat-a-count Progesterone; Siemens Diagnostics, USA) and the concentrations of serum estrogens were determined using a direct enzyme-immunoassay (EIA).

Following the manufacturer's instructions, the radioimmunoassay used to measure serum progesterone concentrations, showed an intra-assay coefficient of variation (CV) of 3.1%, and an inter-assay CV of 5.6%. The assay's lower detection limit was 0.03 ng/mL.

For serum estrogen detection, a direct enzyme-immunoassay (EIA) was used. The assay exhibited a recovery range of 84.6 to 96.2%, with intra-assay CV ranging from 1.7 to 13.2%, and an inter-assay CV ranging from 8.2 to 19.7%. The minimal detectable concentration of this assay was 2.5 pg/mL.

Holstein-Friesian heifers underwent 3 cycles of estrus synchronization. The first cycle was devoid of embryo transfer (preconception cycle: PC). The second and third cycles were associated with embryo transfer on Day 7 (1ETD) post-estrus (conception cycles 1 and 2). Females were classified as non-pregnant if pregnancy was undetectable on Day 18 post-estrus at conception cycles 1 and 2. Females were classified as pregnant when the term of pregnancy was reached at conception cycle 1 or when embryo was recovered on Day 18 post-estrus at conception cycles 1 and 2. X: biochemical and/or molecular analyses run with blood sampling.

## PAXgene blood sample collection and processing

Two PAXgene Blood RNA Tubes (BD, Germany; 2.5 ml/tube) were collected per female. For each tube, 2.5 ml of whole blood was drawn from the jugular vein on Day 7 (PCD7) and Day 14 (PCD14) of the preconception cycle, as well as on Day 7 just before the first embryo transfer (1ETD7).

## Total RNA extraction and purification

Total RNA was isolated using the PAXgene Blood RNA system (PreAnalytiX, Qiagen/BD Company, France) following the manufacturer's instructions, with modifications as previously as described [40].

The quality and integrity of the isolated total RNA were assessed using the Agilent 2100 Bioanalyzer (@bridge ICE -Iso Cell Express-, INRA, France: https://www6.jouy.inra.fr/ice). The RNA integrity number (RIN) for each sample ranged from 7 to 9.5.

## Microarray hybridization

Transcriptional profiling was conducted using a custom 8x60K bovine gene expression microarray, designed based on annotated bovine Ensembl transcripts (http://www.ensembl.org/index.html, genome assembly UMD3.1), representing 19,479 transcripts, as previously described [40]. RNA labeling was done with total RNA isolated from blood collected from 7 females classified as non-pregnant and 4 pregnant at PCD7 and PCD14. Due to blood sampling issues, at 1ETD7, microarray hybridization was carried out on total RNA from WBCs collected from 5 non-pregnant and 3 pregnant animals (Table 1). RNA labeling followed the Agilent "One-Color Microarray-Based Gene Expression Analysis" protocol [40]. The microarray data were submitted to the GEO database (accession number GSE185325). Differentially expressed genes (DEGs) were identified using the fold change rank ordering (FCROS) method [41] and t-test with the TREAT option of the LIMMA method [42], comparing non-pregnant with pregnant females. p-values were adjusted using the Benjamini & Hochberg method [43], and a threshold was applied to select significant DEGs.

## Principal component analysis

Principal component analysis (PCA) was performed using the FactoMineR package [44] to assess sample dispersion, first using log2-transformed counts of all genes, then only the DEGs. Confidence ellipses around categories represent the means with a 0.95 confidence level, based on empirical variance divided by the number of observations.

## Ingenuity pathway and interaction network analysis

Ingenuity pathway analysis was performed to identify associations between DEGs from peripheral white blood cells (WBCs) and canonical pathways, upstream regulators, and gene networks (IPA; Ingenuity® Systems, www.ingenuity.com). The DEGs from PCD7, PDC14, and 1ETD7 were analyzed using a right-tailed Fisher exact test. The analysis included overrepresented canonical pathways, integration of these pathways into biological networks, and prediction of upstream regulators. An interaction network was generated for each DEG list. Additionally canonical pathways and functions were analyzed using the DAVID (Database for Annotation, Visualization and Integrated Discovery) Bioinformatics Resources interface. Functions and pathways with p-values < 0.05 were considered significant.

## Real-time quantitative PCR (qPCR) analysis

For RT-qPCR analysis, additional animals were added, resulting in the following group sizes: animals classified as non-pregnant (n = 9 at PCD7/PCD14, n = 5 at 1ETD7) or pregnant (n = 5 at PCD7/PCD14, n = 3 at 1ETD7). RT-qPCR was performed on 12 selected genes that were differentially expressed in WBCs between the two experimental groups of females.

As previously described [40], first-strand complementary DNA (cDNA) was synthesized from 500 ng of purified total RNA using 0.025 µg Oligo(dT)$_{12-18}$ primers and 200 U/µL of SuperScript II Reverse Transcriptase enzyme (Invitrogen, France) in a 20 µl reaction volume following the manufacturer's protocol. Specific primers for each gene (S13 Table) were designed using NCBI Primer–BLAST [45].

qPCR reactions were prepared with one-quarter of the reverse transcription (RT) reaction volume, 15 µM concentration of primers and 7.5 µl of Sybrgreen Mastermix (Applied Biosystems). Reactions were performed in duplicate in a final volume of 15 µl with RNase/DNase-free water on a StepOnePlus Real-Time PCR Systems (Applied Biosystems, France) following the relative standard curve method [46].

Primers efficiency was validated using standard curves. For each primer pair, a series of six successive dilutions of cDNA was prepared from pooled samples, and then analyzed in duplicate. Amplification curves were generated to calculate the efficiency based on the variation in threshold cycle (Ct) values across the dilution points. The average RT-PCR efficiency was 101.7%.

All PCR reactions were conducted in duplicate. Amplification conditions were as follows: initial denaturation at 95°C for 10 minutes, 45 cycles of denaturation at 95°C for 15 seconds, annealing at 60°C for 60 seconds, and elongation at 72°C for 40 seconds. Product specificity was evaluated by analysing the melting curves and by sequencing of the amplicons. A standard curve was included for each gene to generate arbitrary expression values [46]. Duplicate no-template controls (NTC) were included for each gene to monitor potential contamination. Two plates were used per gene. For the geNorm (Biogazelle) analyses, an inter-run calibration was performed between the two plates for each gene.

Gene expression levels were normalized to three reference genes (*ACTB, GAPDH* and *RLP19*), which showed stable expression across all samples. Reference gene stability was determined using the geNorm module in Qbase software, Biogazelle [47]. Gene expression results are reported as mean calibrated normalized relative quantity (CNRQ) values in arbitrary units.

## Statistical analysis

Statistical analyses of RT-qPCR data were performed using Prism 9.0.0 (GraphPad, USA). Data normality was tested with D'Agostino & Pearson and Shapiro-Wilk tests. Gene expression differences between non-pregnant and pregnant animals at PCD7, PCD14, and 1ETD7 were analyzed using repeated measures two-way ANOVA or mixed-effects model with Geisser-Greenhouse correction, Sidak's multiple comparisons test ($p \leq 0.05$).

## Results

### Pregnancy rates after embryo transfers

All embryos used in this study are from different sires. Upon first embryo transfer (1ET), diagnosis of pregnancy was based on circulating progesterone concentration (P4 > 1 ng/ml or P4 < 1 ng/ml) measured at day 18 post-estrus (Fig 1). Out of 30 Holstein-Friesian heifers, 10 were diagnosed as pregnant. Four of them were left for completing pregnancy to term and uterine flushing was used for the six remaining heifers, leading to the recovery of an elongated conceptus with an embryonic disc. Among the 26 heifers, including 6 initially classified as pregnant and 20 as non-pregnant, 10 conceived after the second embryo transfer, including one that became pregnant following two consecutive transfers. Conversely, 16 heifers remained non-pregnant, with 9 failing to establish pregnancy after both embryo transfers. Accordingly, pregnancy rates (Fig 1) was 33.3% for the first embryo transfer and 36.0% for the second embryo transfer. Analyses were performed on heifers that were allowed to carry pregnancies to term, including one heifer diagnosed as pregnant twice at Day 18 post-embryo transfer, and on heifers that remained non-pregnant after two consecutive rounds of embryo transfer. These groups were subsequently referred to as "pregnant" and "non-pregnant" females in the following analyses (S1 Table).

### Changes in serum steroid hormones

On Days 7 and 14 post-estrus of the preconception cycle (PCD7, PCD14), circulating progesterone concentrations (P4) were significantly higher in non-pregnant heifers (n = 7) than in the pregnant 4 heifers (Fig 2A). Upon embryo transfer, P4 concentrations were significantly higher in pregnant heifers than non-pregnant animals at Day 18 (Fig 2C). Estrogens concentrations did not differ significantly between the groups (Fig 2B and 2D).

### Transcriptomic profiling and analysis

Using a custom bovine gene expression array containing 19,479 unique transcripts- element bovine oligonucleotide array [40], gene expression profile were determined using WBCs collected at Day 7 post-estrus (PCD7) and Day 14

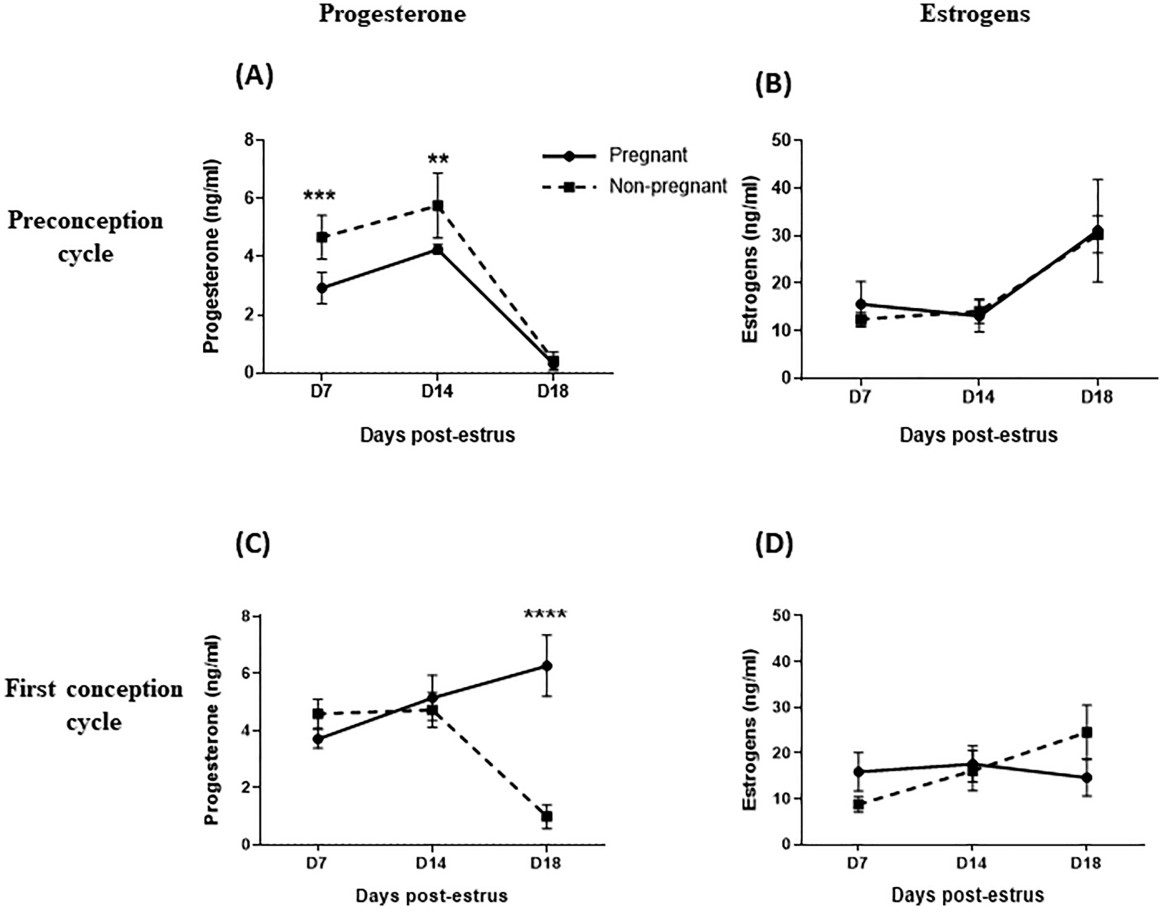

**Fig 2. Progesterone and estrogens concentrations (ng/ml) on days 7, 14 and 18 post-estrus in Holstein-Friesian heifers.** Blood was collected from Holstein-Friesian heifers, which were subsequently classified as pregnant (n = 4) or non-pregnant (n = 9) based on pregnancy diagnosis, determined by circulating P4 levels and embryo recovery on Day 18 post-embryo transfer. Females were sampled on Days 7, 14 and 18 post-estrus during the preconception cycle **(A, B)** and the first conception cycle **(C, D)**.

post-estrus (PCD14) of the preconception cycle as well as Day 7 of the cycle associated with embryo transfer (1ETD7). Based on RNA samples, transcriptomes were analyzed in 7 heifers classified as non-pregnant and 4 heifers classified as pregnant at PCD7 and PCD14 as well as 5 out of 7 heifers of the non-pregnant group and 3 out of the 4 females of the pregnant group at 1ETD7. The fourth heifer, diagnosed as pregnant after the first embryo transfer and allowed to carry the pregnancy to term, was excluded from the analysis due to blood collection occurring after the embryo transfer. Therefore, the heifer that became pregnant following two successive embryo transfers was included in the pregnant group (S1 Table). Volcano plots that combined fold change (FC) and probabilities (f-value ≤ 0.015 for low-expressed probes and f-value ≥ 0.985 for over-expressed probes) from the fold change rank ordering statistics (FCROS) method [41] were used to visualize DEGs distribution between females of the non-pregnant group and of the pregnant group at Days PCD7, PCD14 and 1ETD7 (Fig 3A–C). The low-expressed and over-expressed probes were indicated in green and red color respectively. Using TREAT- option of the LIMMA statistical method (t-test relative to a threshold, p-value < 0.05 and FDR ≥ 10%; FC ≥ 1.2 or FC ≤ 0.80), we identified 1240 DEGs, 896 DEGs and 1023 DEGs between the females classified as non-pregnant and those classified pregnant group at PCD7, PDC14 and 1ETD7 respectively.

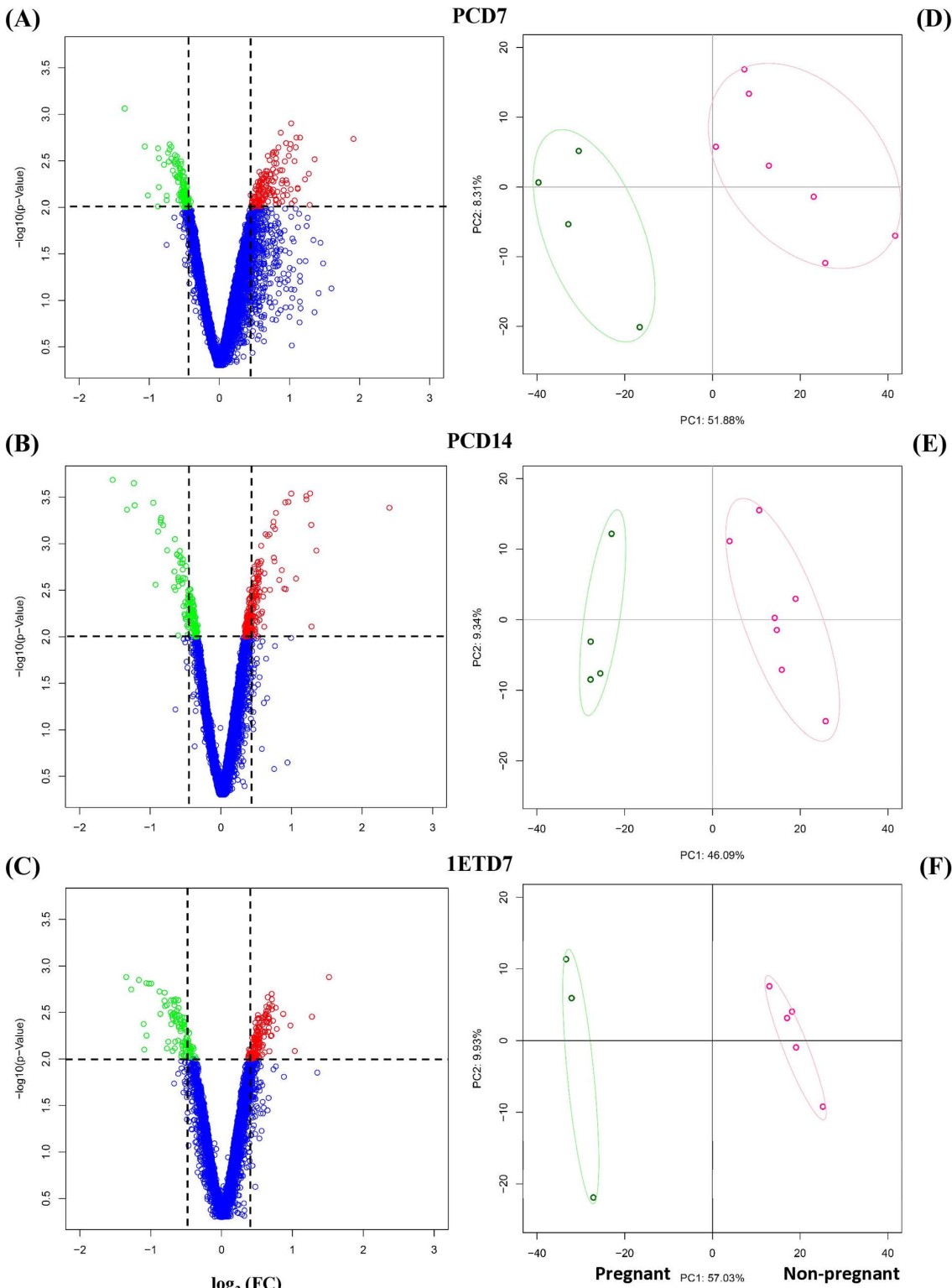

**Fig 3. Volcano plots and Principal component analysis (PCA) showing the association between gene expression in peripheral white blood cells (WBCs) and physiological status of Holstein-Friesian heifers. (A, B, C)** Volcano plots of gene expression profiles (all probes) in peripheral white blood cells on Day 7 (PCD7) and 14 (PCD14) of the preconception cycle comparing heifers based on pregnancy outcome at first conception

(classified as non-pregnant, n = 7; pregnant, n = 4), as well as on Day 7 of the conception cycle (1ETD7) at the time of embryo transfer (classified as non-pregnant, n = 5; pregnant, n = 3). Each dot represents the mean expression of an individual gene obtained from a microarray normalized dataset. The cut-off values (black dotted lines) were established according to the following parameters: |log10(fold change (FC)| ≥ 2. and FDR ≥ 10%. The red dots represent the over-expressed transcripts (f.value ≤ 0.05) and the green dots represent the under-expressed transcripts (f.value ≥ 0.95). Genes above the cut-off lines have been considered as differentially expressed. The y-axis corresponds to the mean expression value of log 10 (p-value), and the x-axis displays the log2 fold change value (females classified as non-pregnant vs. females classified as pregnant at D18 post-estrus following embryo transfer). (D, E, F) Principal Component Analysis (PCA) of differentially expressed genes (DEGs) from microarray data of peripheral white blood cells (WBCs) obtained from females classified as non-pregnant (n = 7) and pregnant (n = 4) at PCD7, PCD14, and 1ETD7 (classified as non-pregnant, n = 5; pregnant, n = 3), with a p-value ≤ 0.05.

At PCD7, expression levels of 550 DEGs and 690 DEGs were higher and lower respectively in WBCs from females classified as non-pregnant group when compared with females of the pregnant group (S2 Table). At PCD14, expression levels of 417 DEGs and 479 DEGs were higher and lower respectively in the WBCs from females of the non-pregnant group when compared with females of the pregnant group (S3 Table). At 1ETD7, expression levels of 532 DEGs were higher and 491 DEGs were lower in WBCs from females qualified as non-pregnant compared to pregnant (S4 Table). Based on the DEGs, principal component analyses showed a separation between the WBCs transcriptomic profiles of the non-pregnant and pregnant animals at PCD7, PCD14 or 1ETD7 (Fig 3D–F).

### Pregnancy success is associated with distinct gene expression profiles in white blood cells (WBCs)

Venn diagram was used to compare the numbers of genes differentially expressed in WBCs isolated from females classified as non-pregnant and pregnant (Fig 4). Using TREAT, we found that 998 DEGs (FC ≥ 2, n = 8), 655 DEGs (FC ≥ 2, n = 9) and 847 DEGs (FC ≥ 2, n = 15) were specific to PCD7, PCD14, and 1ETD7, respectively. Whereas 132 DEGs were common to PCD7 and PCD14, 77 DEGs were common to PCD7 and 1ETD7, and 76 DEGs were common to PCD14 and 1ETD7. A total of 33 DEGs were found to be common between PCD7, PCD14 and 1ETD7 (Fig 4 and S5 Table).

The heatmaps generated from the Top 100 DEGs in WBCs from the heifer group that became pregnant and those that were unable to conceive revealed that gene expression patterns were correlated with both the stage of the estrous cycle (PCD7, PCD14, and 1ETD7) and the animals' phenotype (Fig 4). These DEGs are linked to various biological pathways, including immune response and inflammation, cell proliferation and cycle regulation, lipid and phospholipid metabolism, skeletal development and mineralization, as well as development, differentiation, apoptosis, and cell death regulation.

Among the 33 DEGs common to PCD7, PCD14 and 1ETD7, 16 DEGs consistently presented higher expression levels (FC ≥ 1.2 or FC ≤ 0.80, p-value < 0.05) in the WBCs of the heifers classified as non-pregnant, while 10 DEGs consistently exhibited higher expression level in the heifers classified as pregnant. The expression levels of 7 DEGs were variable across PCD7, PCD14 and 1ETD7 (Fig 5 and S5 Table).

### Networks associated with differentially expressed genes (DEGs) in WBCs related to pregnancy outcome

The differentially expressed WBC genes between heifers classified as non-pregnant and pregnant were mapped onto a global molecular network using Ingenuity Pathways Analysis (IPA). At PCD7, the IPA network module identified 25 networks, with one of the main networks, "Cell Signaling, Nucleic Acid Metabolism, and Small Molecule Biochemistry" (n = 24 genes, 22 of which are focus genes), linking genes involved in biological functions such as cell signaling, nucleic acid metabolism, and small molecule biochemistry, with a central role played by a G protein-coupled receptor (Fig 6A and S6 Table) in the non-pregnant group. At PCD14, 25 networks were also identified, with the most biologically relevant network related to "Developmental Disorder, Organismal Injury and Abnormalities, Renal Hypoplasia" (n = 28 genes, 23 of which are focus genes, Fig 6B and S7 Table) in the non-pregnant group. This network links genes involved in biological processes that influence organ structure and function. At 1ETD7, among the 25 networks identified, the main network was associated with "Neurological Disease, Cellular Growth and Proliferation, Hematopoiesis" (n = 41 genes, 30 of which are

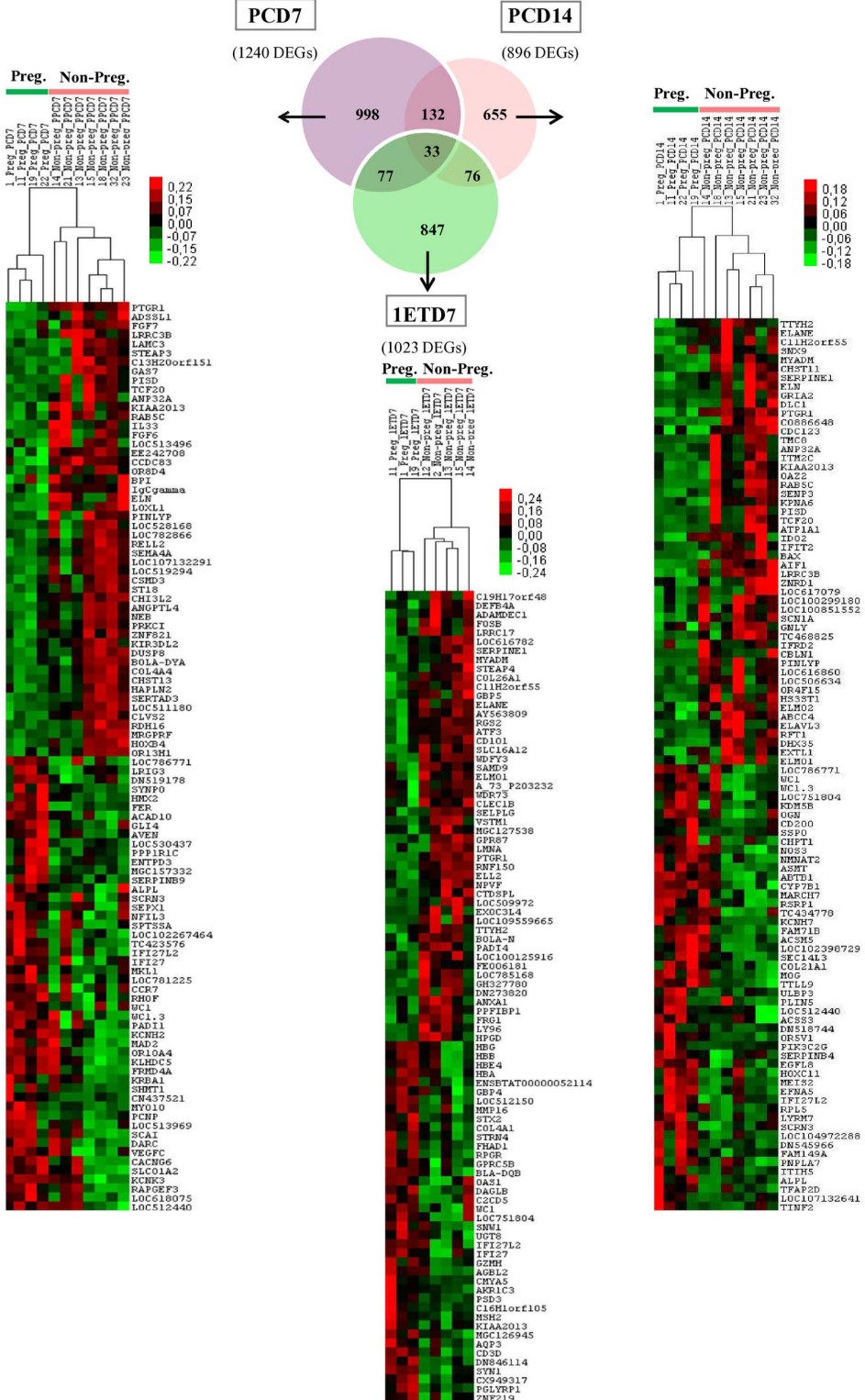

**Fig 4. Venn diagrams and Heatmaps of differential gene expression in peripheral white blood cells (WBCs) of Holstein-Friesian heifers according to the ability to become pregnant.** Heatmaps highlighting the Top 100 differentially expressed genes (DEGs), with 50 overexpressed and 50 underexpressed, identified in WBCs on Days 7 (PCD7) and 14 (PCD14) of the preconception cycle, stratified by pregnancy outcome at the first

conception cycle (classified as non-pregnant, n = 7, pregnant, n = 4), as well as at Day 7 of the conception cycle at the time of embryo transfer (classified as non-pregnant, n = 5, pregnant; n = 3). Heatmaps were generated using Cluster 3.2 software and visualized with TheeView software. P-value ≤ 0.05.

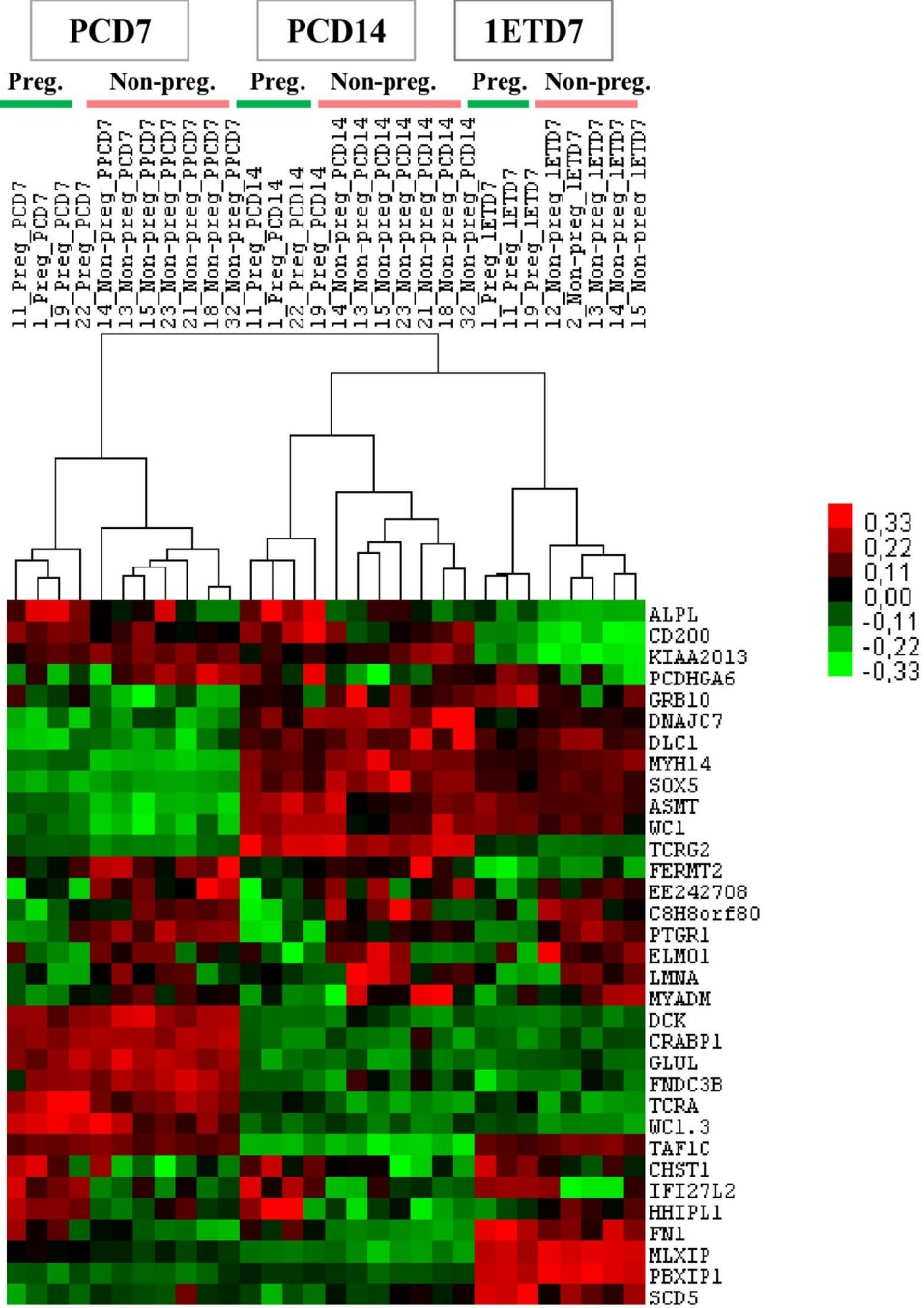

**Fig 5. Heatmap highlighting 33 common differentially expressed genes (DEGs) at PDC7, PCD14 and 1ETD7.** DEGs were identified in peripheral white blood cells (WBCs) from Holstein-Friesian heifers, classified as non-pregnant (Non-preg.) and pregnant (Preg.). Samples were collected on Days 7 and 14 of the preconception cycle, as well as on Day 7 of the first conception cycle prior to embryo transfer. A heatmap was generated using Cluster 3.2 software and visualized with TreeView software.

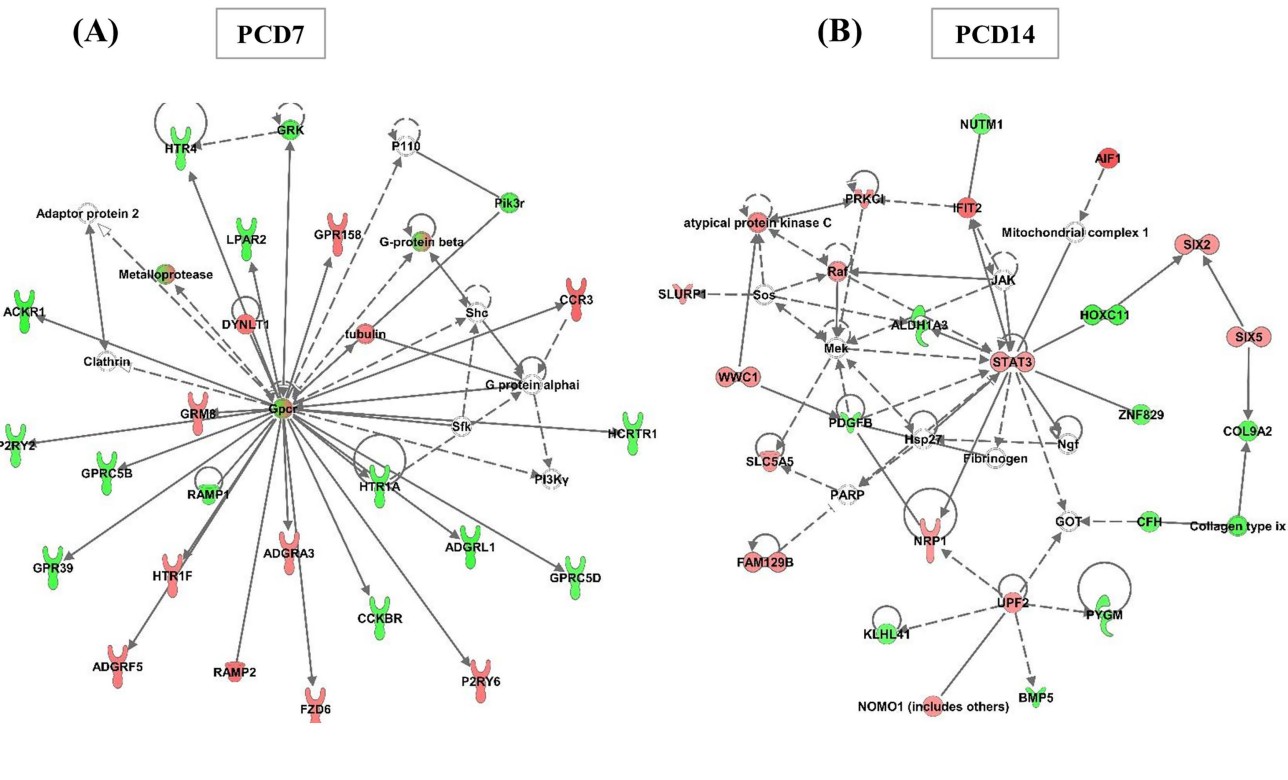

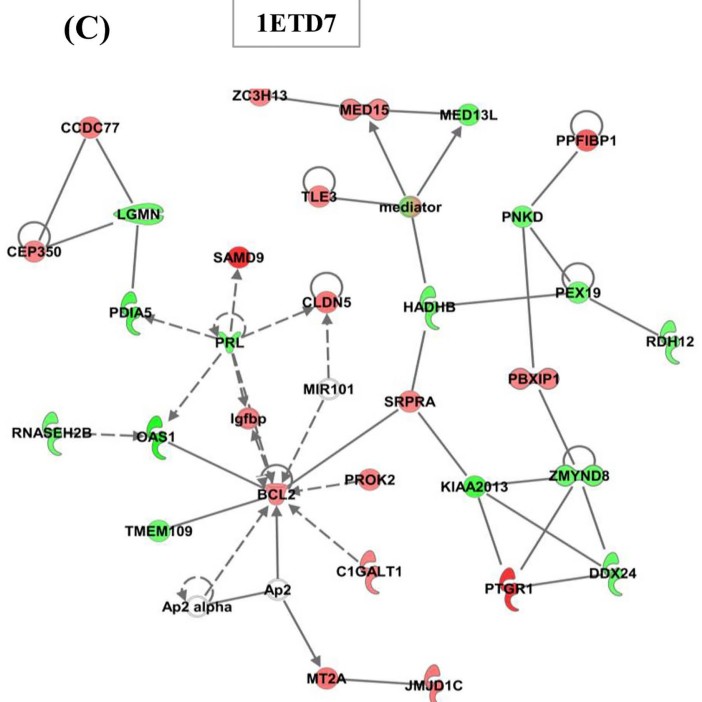

**Fig 6. Major networks of transcripts that are associated with the ability of Holstein-Friesian heifers to become pregnant upon embryo transfer.**
Selected networks (based on) contain overexpressed and underexpressed differentially expressed genes (DEGs) identified in peripheral white blood cells (WBCs) collected from the two groups of Holstein-Friesian heifers on Days 7 and 14 of the preconception cycle (PCD7 and PCD14), as well as on Day 7 of the first conception cycle prior to embryo-transfer (1ETD7).

focus genes, Fig 6C and S8 Table) in the non-pregnant group. This network connects genes that regulate nervous system function, orchestrate hormonal and organ responses, and control cell proliferation critical for organ development, preparation, hematopoiesis, and immune function. Ingenuity Pathways Analysis also revealed a central role for fibronectin 1 (FN1) and ubiquitin B (UBB) in organizing this network.

Top functions in the selected networks are linked to (A) "Cell Signaling, Nucleic Acid Metabolism, Small Molecule Biochemistry" at PCD7; (B) "Developmental Disorder, Organismal Injury and Abnormalities, Renal Hypoplasia" at PCD14, and (C) "Neurological Disease, Cellular Growth and Proliferation, Hematopoiesis" at 1ETD7. Network displays nodes (genes/gene products) and edges (the biological relationship between nodes). Red and green shaded nodes represent up- (overexpressed) and down-regulated (underexpressed) gene expression, respectively. The color intensity of the nodes indicates the fold change increase (red) or decrease (green) associated with a particular gene. Solid line indicates a direct interaction between nodes (genes/gene products) and a dashed line indicates an indirect relationship between nodes. White symbols indicate neighboring genes that are functionally associated, but not included in the differentially expressed gene list. The shape of the node is indicative of its function (legend available online https://www.qiagenbioinformatics.com/products/ingenuity-pathway-analysis).

### Biological functions and canonical pathways linked with differentially expressed genes in WBCs associated with pregnancy outcome

Biological terms and functions associated with differentially expressed WBC genes between females classified as non-pregnant and pregnant were identified using DAVID. Gene Ontology (GO) and Kyoto Encyclopedia of Genes and Genomes (KEGG) analyses were performed on 1240, 896, and 1023 DEGs identified at PCD7, PCD14 and 1ETD7, respectively. The top five significant terms/functions were presented for each time point in Fig 7. A total of 27, 11, and 28 functional categories and canonical pathways c (n ≥ 5 genes by category, p-value ≤ 0.05) were associated with DEGs at PCD7, PCD14, and 1ETD7 respectively (S9 Table).

The major functional category terms and canonical pathways associated with highly expressed genes in heifers classified as non-pregnant were "regulation of growth" and "leukocyte transendothelial migration" at PCD7, "inflammatory response" and "endocytosis and inflammatory response" at PDC14, and "angiogenesis" and "HTLV-I infection" at 1ETD7, respectively (Fig 7). In heifers classified as pregnant, DEGs were associated with "microtubule cytoskeleton organization" and "Calcium signaling pathway" at PCD7, "nucleic acid binding and positive regulation of angiogenesis" at PCD14, and "GTPase activity and metabolic pathways" at 1ETD7 (Fig 7).

### Upstream regulators linked with differentially expressed genes in WBCs associated with pregnancy outcome

The upstream analysis module of IPA identified 694, 1,087, and 1,401 upstream regulators (p-value of overlap ≤ 0.05) at PCD7, PCD14, and 1ETD7, respectively (S10–S12 Tables), in WBCs collected from heifers classified as non-pregnant compared to those classified as pregnant. Among these, 142 upstream regulators were common across all three-time points (S13 Table), with 24 showing significant activation or inhibition (Z-score ≥ +2 or ≤ −2, p-value of overlap ≤ 0.05; Table 2). Common upstream regulators, including the growth factor AGT, the transcription factor EGR2, the cytokines IFNG and IL1B, and the chemical compound lipopolysaccharide, which are involved in immune and inflammatory responses, were predicted to be activated.

Conversely, the chemical kinase inhibitor PD98059, which is implicated in proliferation, differentiation, and survival; the growth factors VEGF; and VEGFA, involved in angiogenesis and vascular permeability; and the mature microRNAs miR-155-5p (with the UAAUGCU seed sequence) and miR-16-5p (along with other miRNAs sharing the AGCAGCA seed sequence), which regulate immune responses and inflammation, were predicted to be inhibited (Table 2).

Among the 376 upstream regulators specific to PCD7, the chemical drug (indomethacin); the calcium-binding protein A8 (S100A8) and the transcription regulator (GATA6) were activated, while the mature microRNA (miR-146a-5p (and other

# Gene Ontology (GO) and KEGG Pathway enrichment at PCD7, PCD14, and 1ETD7

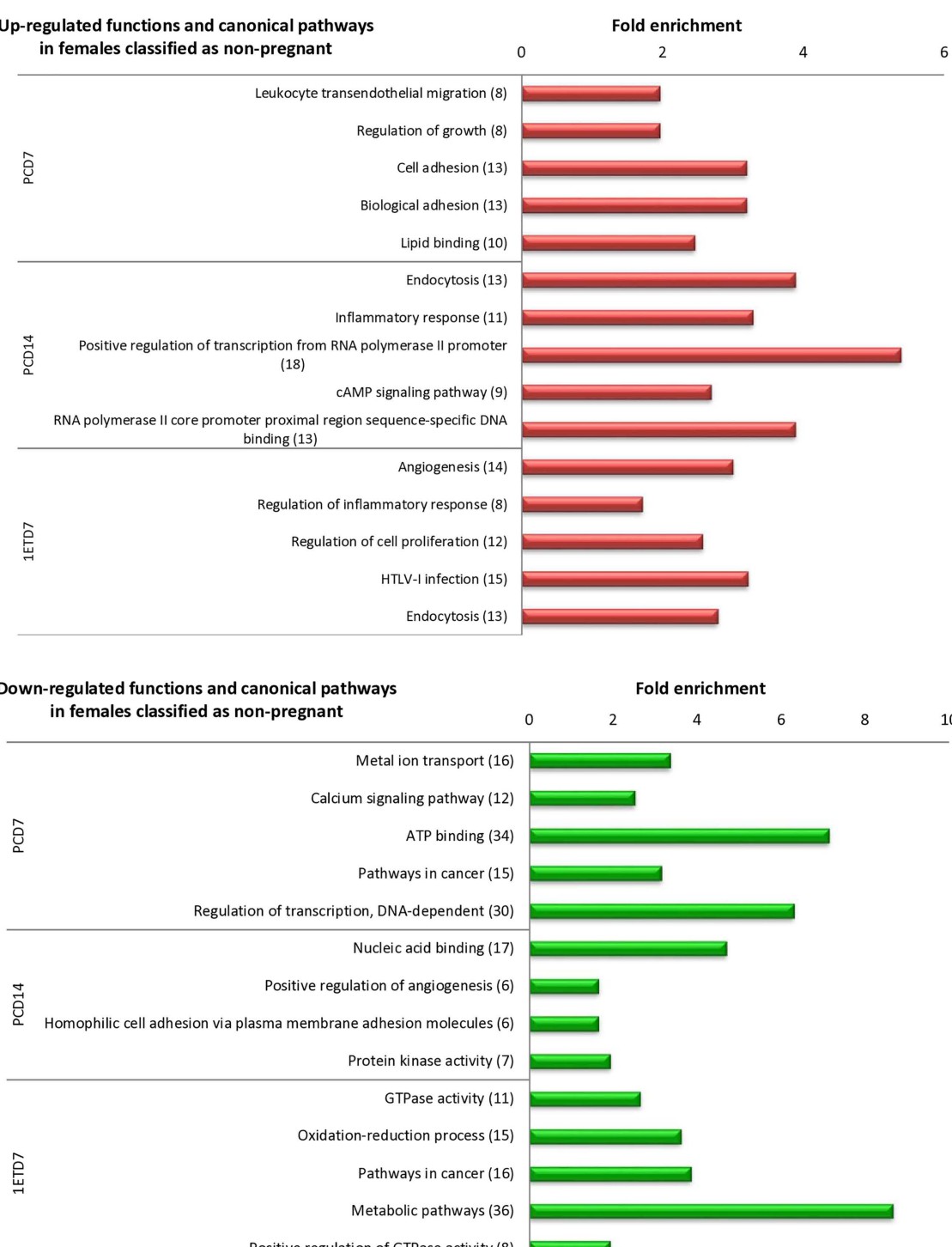

**Fig 7. Top five functional categories (GO and KEGG) of genes significantly enriched in peripheral white blood cells collected from Holstein-Friesian heifers.** Using DAVID tools, analyses were conducted on significantly (P<0.05) up- (overexpressed) or underexpressed genes identified between heifers that were classified as non-pregnant (n=7) vs. pregnant (n=4) on Days 7 of the preconception cycle (PCD7, PCD14), as well as

on Day 7 of the first conception cycle prior to embryo transfer (1ETD7). Bars represent the fold enrichment and number of genes is indicated in brackets. The list of genes under each category is provided in S9 Table.

**Table 2. Top activated and inhibited upstream regulators in peripheral white blood cells (WBCs) common to the preconception cycle days (Days 7 and 14 post-estrus: PCD7 and PCD14) and Day 7 of the first conception cycle prior to embryo transfer (1ETD7).**

| Upstream Regulator | Molecule Type | Predicted Activation State | PCD7 | | PCD14 | | 1ETD7 | |
|---|---|---|---|---|---|---|---|---|
| | | | Activation z-score | p-value of overlap | Activation z-score | p-value of overlap | Activation z-score | p-value of overlap |
| AGT | growth factor | Activated | 3.63 | 4.26E-05 | 3.13 | 3.28E-04 | 4.31 | 1.55E-04 |
| CD38 | enzyme | Activated | 2.77 | 1.31E-02 | 2.43 | 3.59E-02 | 2.81 | 1.50E-02 |
| Cigarette smoke | chemical toxicant | Activated | 2.40 | 8.47E-04 | 2.22 | 2.26E-02 | 2.77 | 3.05E-03 |
| cyclic AMP | chemical – endogenous mammalian | Activated | 2.37 | 2.46E-03 | 2.24 | 2.26E-02 | 2.13 | 3.05E-03 |
| EGR2 | transcription regulator | Activated | 2.18 | 4.31E-03 | 2.77 | 3.31E-03 | 2.47 | 1.48E-03 |
| F2 | peptidase | Activated | 3.71 | 1.04E-03 | 3.22 | 3.44E-03 | 3.28 | 4.44E-04 |
| HGF | growth factor | Activated | 3.73 | 1.82E-02 | 2.89 | 4.22E-03 | 3.15 | 1.14E-02 |
| Hydrogen peroxide | chemical – endogenous mammalian | Activated | 3.88 | 1.58E-02 | 3.79 | 3.21E-03 | 2.61 | 4.90E-03 |
| IFNG | cytokine | Activated | 2.92 | 2.15E-04 | 4.52 | 7.41E-05 | 3.30 | 1.54E-05 |
| IL1A | cytokine | Activated | 3.09 | 2.44E-02 | 2.08 | 3.76E-02 | 2.74 | 4.39E-05 |
| IL1B | cytokine | Activated | 2.47 | 1.25E-03 | 2.06 | 1.19E-03 | 3.03 | 1.17E-06 |
| IL2 | cytokine | Activated | 2.75 | 4.31E-03 | 3.59 | 3.57E-03 | 3.16 | 2.23E-05 |
| IL5 | cytokine | Activated | 3.22 | 2.59E-02 | 3.14 | 1.32E-02 | 3.38 | 1.33E-02 |
| Lipopolysaccharide | chemical drug | Activated | 3.72 | 5.08E-03 | 4.22 | 2.81E-04 | 6.07 | 8.69E-12 |
| miR-155-5p (miRNAs w/ seed UAAUGCU) | mature microRNA | Inhibited | −2.92 | 5.04E-03 | −2.81 | 3.84E-03 | −2.76 | 1.77E-02 |
| miR-16-5p (and other miRNAs w/seed AGCAGCA) | mature microRNA | Inhibited | −2.91 | 2.36E-02 | −2.78 | 1.65E-02 | −3.09 | 1.05E-02 |
| NFkB (complex) | complex | Activated | 2.82 | 8.40E-03 | 2.98 | 2.13E-02 | 4.12 | 1.35E-03 |
| PD98059 | chemical – kinase inhibitor | Inhibited | −3.10 | 3.09E-05 | −2.14 | 2.53E-04 | −3.75 | 4.75E-05 |
| Tetradecanoylphorbol acetate | chemical drug | Activated | 3.77 | 2.91E-02 | 3.25 | 3.91E-04 | 3.07 | 3.74E-06 |
| TGFB1 | growth factor | Activated | 5.70 | 1.73E-04 | 3.77 | 7.26E-06 | 3.95 | 1.96E-04 |
| TNF | cytokine | Activated | 3.77 | 2.91E-02 | 2.55 | 5.38E-05 | 4.21 | 2.17E-06 |
| U0126 | chemical drug | Inhibited | −3.36 | 6.06E-04 | −2.70 | 5.22E-06 | −2.27 | 8.49E-04 |
| VEGF | group | Activated | −2.24 | 1.89E-04 | 2.70 | 4.63E-07 | 2.94 | 8.52E-03 |
| VEGFA | growth factor | Activated | −3.36 | 6.06E-04 | 2.19 | 8.07E-04 | 2.48 | 8.07E-03 |

miRNAs w/seed GAGAACU)), the chemical – protease inhibitor (N-[N-(3,5-difluorophenacetyl-L-Ala)]-S-phenylglycine t-butyl ester) and the transcription regulator (TAF4) were inhibited.

Among the 558 upstream regulators specific to PCD14, the transcription regulators (TP63, POU5F1) and the ligand-dependent nuclear receptor (AR) were activated, while fusion gene/product (ETV6-RUNX1), the enzyme (EGLN) and the ion channel (GRIN3A) were inhibited. Among the 896 upstream regulators specific to 1ETD7, the transcription regulator (STAT4) and the cytokines (PRL, IL1), were activated, while the transcription regulator (CITED2), the signaling regulator (TSC2), and the phosphatase (PTPRR) were inhibited.

## Validation of selected genes in WBCs by RT-qPCR relatively to pregnancy outcome

Twelve [12] WBC-expressed genes (*AIF1, DCK, FNDC3B, GLUL, IL15, IFI27L2, MYADM, PTGR1, SERPINE1, SOX5, TCN2* and *VASH1*) were selected (Table 3) and their expression was quantified using RT-qPCR.

These genes were selected based on their differential expression between heifers classified as non-pregnant and those classified as pregnant at each time point, with two additional heifers included in the non-pregnant group (n = 9). Of the 12 candidate genes, 6 have been validated by RT-qPCR. *PTGR1* (Prostaglandin reductase 1, FC = 4.8) mRNA abundance was significantly (p ≤ 0.05) greater in WBCs from the non-pregnant group compared to those from the pregnant group at PCD7, PCD14, and 1ETD7 (Fig 8). *GLUL* (FC = 1.34) transcript was significantly higher (p ≤ 0.05) at PCD7 or exhibited a trend toward a higher expression (p = 0.054) in the heifers of the non-pregnant group than the heifers of the pregnant group at PCD14, (Fig 8). *AIF1* (FC = 1.99), and *SERPINE1 (FC = 1.76),* mRNA levels were significantly higher (p ≤ 0.05) or exhibited a trend toward a higher expression (p = 0.14) in the heifers of the non-pregnant group at PCD14, respectively

**Table 3. Primer sequences used for real-time PCR (RT-qPCR) quantification of transcript levels of a selection of 12 differentially expressed genes (DEGs) and 3 reference genes.**

| Gene Name | Unigene/ GeneID | Accession | Forward (For.) and reverse (Rev.) primers sequences (5'-> 3) | Product length (bp) |
|---|---|---|---|---|
| *AIF1* | ID:280989 | NM_173985 | For:TCTGCCATTCTAAAAATGATCCTG | |
| | | | Rev:CCTCACCACAAATCAGGGCA | 114 |
| *DCK* | ID:530642 | NM_001034573 | For:GAGCCACTCCAGAGAAATGCTT | |
| | | | Rev:TTCTATGCAGGAGCCAGCTTT | 123 |
| *FNDC3B* | ID:615534 | NM_001206211 | For:GAGGAGACCCTCACTGCCTA | |
| | | | Rev:GGCCTGAATTCTGATCCGGT | 198 |
| *GLUL* | ID:281199 | NM_001040474 | For:GGTTAACAGGTGGAAGCCCA | |
| | | | Rev:GGCCTGCTTTAGTGACATGC | 184 |
| *IFI27L2* | ID:535465 | NM_001076061 | For:TCCAGTCCGTGGGGGC | |
| | | | Rev:TGGAGGAGGGGAAGGAGATG | 119 |
| *IL15* | ID:281248 | NM_174090 | For:CAGCGATGCAGTGCTTTCTC | |
| | | | Rev:TCCTCACATTCTTTGCATCCCA | 157 |
| *MYADM* | ID:506295 | NM_001075252 | For:CTGACAGGCATCAACCTGCT | |
| | | | Rev:TGAGGGACACAGACAAGGGA | 228 |
| *PTGR1* | ID:513177 | NM_001035281 | For:ATCGCTAAGCTCAAGGGCTG | |
| | | | Rev:CGGGAGCAGCTTCTTTCAGA | 145 |
| *SERPINE1* | ID:281375 | NM_174137 | For:CTCCGAGGTTTGAGACCCAC | |
| | | | Rev:ACACCTTGTCTCTCCTTGCG | 157 |
| *SOX5* | ID:533829 | NM_001083471 | For:ATAGAGAATCCAGGGGGCGT | |
| | | | Rev:CTTTCCAGCGAGATCCCAGT | 159 |
| *TCN2* | ID:281518 | NM_174195 | For:CTCCAGACACCCCGGTAATG | |
| | | | Rev:TGAGTGACCTTCTCGGGACA | 203 |
| *VASH1* | ID:615419 | NM_001206803 | For:AGACCTCTGACAGGGCTGAT | |
| | | | Rev:CACGATGTGGCGGAAGTAGT | 177 |
| **Reference genes** | | | | |
| *ACTB* | ID:280979 | NM_173979.3 | For:GAATCCTGCGGCATTCACGA | 186 |
| | | | Rev:GCGCGATGATCTTGATCTTCATT | |
| *GAPDH* | ID:281181 | NM_001034034.2 | For:GCCGATGCCCCCATGTTTGT | 150 |
| | | | Rev:TCATAAGTCCCTCCACGATGC | |
| *RPL19* | ID:510615 | NM_001040516.2 | For:CCCAATGAGACCAATGAAATC | 73 |
| | | | Rev:CAGCCCATCTTTGATCAGCTT | |

(Fig 8). The mRNA expression of *FNDC3B* (FC = 1.31), *IL15 (FC = 1.44),* and *SERPINE1* were significantly higher (p ≤ 0.05) or exhibited a trend toward a higher expression (p = 0.06) in the heifers of the non-pregnant group at 1EDT7, respectively (Fig 8).

On Day 7 (PCD7) and Day 14 (PCD14) of the preconception cycle, as well as on Day 7 of the first conception cycle prior to embryo transfer (1ETD7), expression levels of *AIF1, FNDC3B, GLUL, IL15, PTGR1,* and *SERPINE1* transcripts were measured by RT-qPCR in females classified as non-pregnant (n = 9) and pregnant (n = 4).

The horizontal line denotes the median value, the box encompasses the upper and lower quartiles, whiskers show the range, and the plus symbol denotes the mean. A Repeated Measures (RM) two-way ANOVA or Mixed-effects Model with Geisser-Greenhouse correction was applied, with matched values stacked into a subcolumn. Sidak's multiple comparisons test was performed, with individual variances computed for each comparison. The results of this test are denoted in the plot with horizontal bars and asterisks (*P < 0.05; ** P < 0.01).

## Discussion

The establishment of a successful pregnancy following embryo transfer depends on reciprocal and synchronized interactions between the competent developing embryo and the maternal endometrium. The quality of an embryo, defined by its potential to develop to term, has been extensively studied using robust and relevant parameters [48,49]. Similarly, predicting the capacity of recipient females to support a pregnancy to term, resulting in the birth of healthy offspring, has been a subject of research. In cattle, omics-based phenotyping of endometrial biopsies has identified promising biomarkers, but their routine application remains challenging [6,9,50,51]. Consequently, identifying minimally invasive biomarkers to reliably predict pregnancy potential in recipient females without endometrial tissue sampling represents a critical challenge with significant commercial applications in reproductive management.

The present study, compared two groups of Holstein-Friesian heifers: (i) those that successfully carried a pregnancy to term after the first embryo transfer following an initial preconception cycle, including one heifer that was confirmed pregnant twice on Day 18, all classified as "pregnant"; and (ii) heifers that failed to establish pregnancy after two consecutive embryo transfers, classified as "non-pregnant". Compared with heifers classified as pregnant, heifers classified as non-pregnant heifers exhibited higher circulating progesterone (P4) concentrations on Days 7 and 14 post-estrus in the preconception cycle, consistent with previous study that showed a relationship between elevated plasma P4 levels and decreased conception rates in cows [52]. Progesterone levels alone are insufficient to predict capacity of pregnancy and pregnancy outcomes, underscoring the need to explore additional markers. Peripheral white blood cells (WBCs), which secrete various factors in response to steroid hormones, represent a potential source of maternal biomarkers for predicting gestational success [53]. Human studies suggests that peripheral lymphocyte balance, cytokine profiles, and specific interleukins can serve as predictive biomarkers of implantation before embryo transfer [54,55]. However, to our knowledge, this approach has not yet been investigated in cattle in the context of embryo transfer.

Our transcriptomic analyses reveal distinct gene expression profiles in WBCs during the estrous cycle and at the time of embryo transfer, which are associated with implantation potential and pregnancy outcomes. WBCs of heifers classified as non-pregnant exhibited overexpression of genes linked to immune and inflammatory responses, suggesting exacerbated inflammation and immune activation, whereas, heifers classified as pregnant showed a regulated innate immune response supporting pregnancy maintenance. Heifers categorized as non-pregnant displayed elevated expression of pro-inflammatory cytokine gene including *IL15,* chronic inflammation-associated genes (*SERPINE1, AIF1*), and immune response gene (*FNDC3B*). These data suggest that immune dysregulation during the estrous cycle may impair uterine receptivity and hinder pregnancy establishment in Holstein-Friesian heifers. This aligns with a former study showing that a dynamic network of cytokines tightly regulates pregnancy initiation and maintenance in women. Pro-inflammatory cytokines such as TNF-α and IL-1β play critical roles during implantation and early placentation, while anti-inflammatory cytokines such as IL-10 and TGF-β are essential for immune tolerance and fetal development. The authors further showed

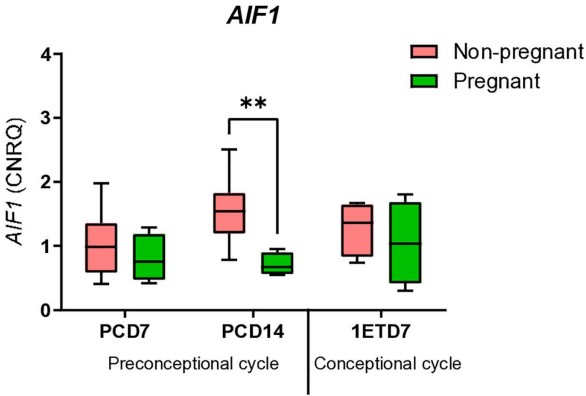

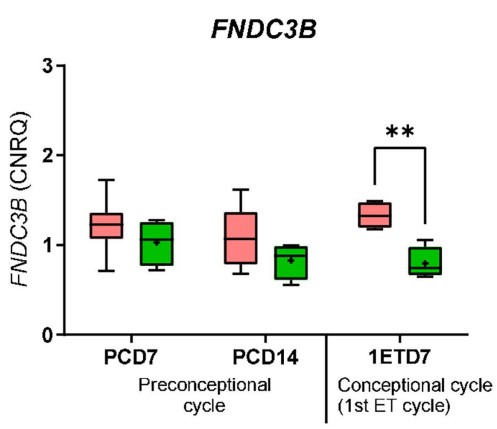

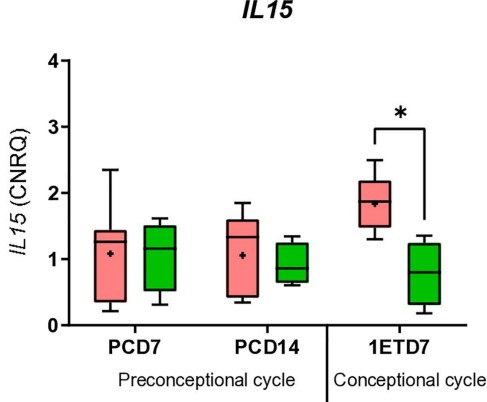

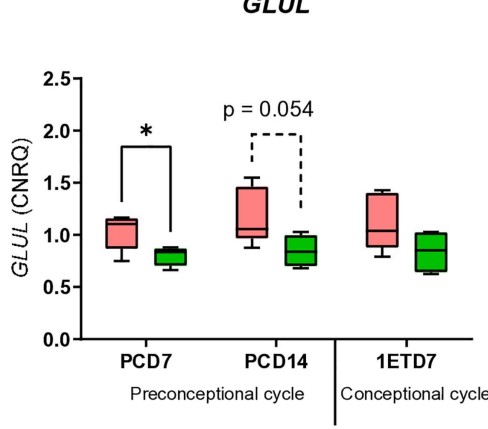

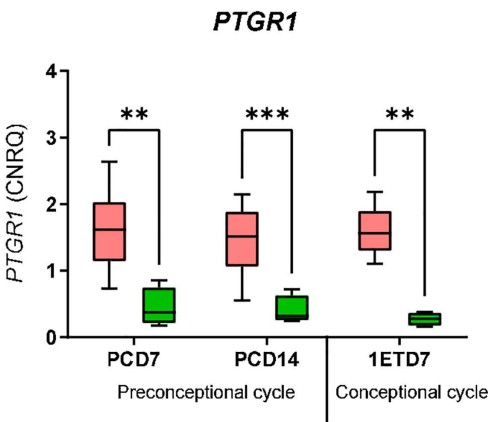

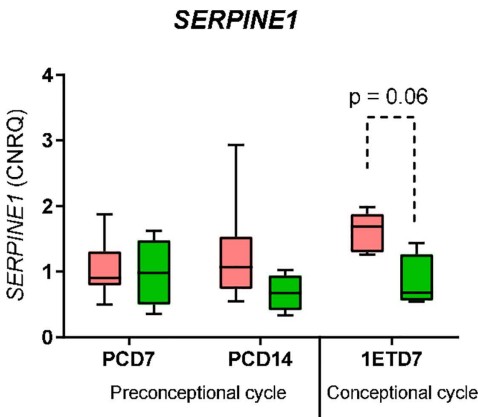

**Fig 8. Gene expression in peripheral white blood cells (WBCs) of Holstein-Friesian heifers on Days 7 and 14 of the preconception cycle (PCD7 and PCD14), and on Day 7 of the first conception cycle (1ETD7) prior to embryo transfer.**

that dysregulation of this cytokine balance is linked with miscarriage and pregnancy complications [56]. The IPA analysis of our study showed that the cytokines TNF-α and IL-1β are predicted to be upstream regulators activated in the heifers classified as non-pregnant.

A significant overexpression of Allograft Inflammatory Factor 1 (*AIF1*) transcripts was observed in WBCs of heifers classified as non-pregnant on Day 14 post-estrus of the preconception cycle. AIF1, a cytokine implicated in chronic allograft rejection in the rat model [57], uterine immune response in dog [58], and inflammatory responses in peripheral blood monocytes and synovial membranes in humans [59], may indicate an inadequate or suboptimal maternal systemic environment for embryo implantation and pregnancy progression. Similarly, *IL15* transcripts were significantly elevated at the time of embryo transfer in WBCs from heifers classified as non-pregnant. This finding is consistent with human data associating high IL-15 levels with implantation failure and recurrent spontaneous abortions in women [60]. Elevated IL-15, a pro-inflammatory cytokine, may reflect or contribute to the set-up of an immune systemic environment hampering embryo implantation. Exacerbated inflammation has been shown to reduce conception rates in dairy cattle, underscoring the detrimental impact of heightened inflammatory responses on reproductive outcomes [16]. Supporting this hypothesis, a recent study reported that multiparous *Bos indicus*-influenced beef cows with elevated endometrial levels of pro-inflammatory cytokines such as IL-6 and TNF-α showed impaired establishment and maintenance of pregnancy [61].

In the group of non-pregnant heifers, increased expression of *SERPINE1*, which encodes plasminogen activator inhibitor-1 (PAI-1), supports the hypothesis that impaired blood coagulation and fibrinolytic activity may negatively affect pregnancy establishment. In humans, elevated PAI-1 levels are associated with recurrent pregnancy loss and other reproductive disorders [62,63]. Our findings are consistent with a previous work that identified white blood cell genes linked to the infertile phenotype in heifers, with the *SERPINE2* gene believed to play a role in heifer infertility [64].

Interestingly *PTGR1* transcripts were consistently elevated in heifers classified as non-pregnant throughout the preconception cycle and at the day of embryo transfer. PTGR1, a key enzyme in arachidonic acid metabolism, has been shown to play role in inflammation and antioxidant responses [65]. PTGR1 is overexpressed in several cancer cell lines, including hepatocellular carcinoma, lung cancer, prostate cancer and bladder cancer, suggesting an oncogenic function [66]. Although its role in reproduction remains unexplored, the overexpression of *PTGR1* transcripts may reflect a broader dysregulation of inflammatory and antioxidant responses linked to pregnancy failure. Notably, PTGR1 has been identified as a nitroalkene reductase capable of inactivating anti-inflammatory nitro-fatty acids [67]. Therefore, its elevated expression throughout the estrous cycle in the WBCs of heifers classified as non-pregnant may indicate an impaired systemic resolution of inflammation. We hypothesize that this systemic pro-inflammation profile negatively affect endometrial receptivity, ultimately compromising embryo implantation. Further functional studies are needed to better understand the role of *PTGR1* in systemic inflammation and its potential impact on reproductive success in cattle.

WBCs isolated from heifers classified as pregnant exhibited elevated expression of genes related to cell differentiation, lipid metabolism, skeletal mineralization, and embryonic development on Days 7 and 14 post-estrus, both during the preconception cycle and on the day of embryo transfer (Day 7 post-estrus of the conception cycle). These genes included *CLEC1B, COL26A1, MYH14*, and *WDR31*, previously identified as markers of uterine receptivity in dairy cattle endometrium during early diestrus [7]. In addition, genes associated with the major histocompatibility complex (*BOLA-N*), calcium and potassium transport and binding (*CACNG6, KCNH2, S100G*), skeletal mineralization (*ALPL*), actin-based cell shape and motility (*SYNPO, MYO10, MKL1*), cell cycle regulation (*PPP1R1C*), embryonic morphogenesis (*LRIG3*), and cell differentiation and organ development (*KRBA1*) showed increased expression in WBCs from pregnant heifers. These findings support the hypothesis that this gene set could contribute to endometrial receptivity and pregnancy outcomes, possibly through systemic mechanisms. Our results align with elevated transcript expression of solute carrier cluster genes, and genes involved in cell proliferation and the cell division cycle found in the receptive endometrium of heifers on Day 7 of the estrous cycle [8]. Similarly, Mazzoni et al., identified differentially expressed genes related to extracellular

matrix remodeling and cell adhesion in the bovine endometrium during early diestrus, indicating their role in endometrial receptivity and embryo implantation in cattle [7].

Regulatory mechanisms must take place to protect the conceptus from rejection by the maternal immune system [68]. Transcriptomic analyses of endometrial samples collected during the estrous cycle in heifers have demonstrated that genes associated with inflammation, immune responses, and growth factors distinguish between receptive and non-receptive endometrium and are critical for the establishment of pregnancy [8]. Consistent with these findings, before embryo transfer, the endometrium of recipient cows was shown to undergo immune modulation, angiogenesis, and oxidative stress regulation, all key processes that prepare the uterus for pregnancy as early as Day 7 of the estrous cycle [9]. The DEG sets in WBCs varied between PCD7 and 1ETD7, both corresponding to the same estrous day. Notably, candidate genes validated by RT-qPCR showed distinct expression levels between these two stages, indicating that specific transcriptional activity varies between two successive estrous cycles. The gene expression profile in WBCs fluctuates across cycles, and this dynamic variation suggests that gene expression at a specific time point in the estrous cycle may not reliably predict gene expression in the subsequent cycle prior to embryo transfer. This could be explained by the combined influence of hormonal variations during each estrous cycle and the immune system on gene expression in WBCs. Indeed, studies have shown that hormonal fluctuations throughout the estrous cycle lead to dynamic changes in gene expression profiles in the bovine endometrium [4,33]. It can be assumed that these changes might also be reflected in WBCs.

In our study, GO and KEGG pathway analyses revealed the regulation of genes involved in leukocytes transendothelial migration and inflammation response in non-pregnant animals, supporting the hypothesis that peripheral maternal immune modulation facilitates embryo implantation. On the other hand, elevated mRNA expression of pro-inflammatory cytokine genes in WBCs from heifers classified as non-pregnant, compared to pregnant ones, may suggest that the molecular and cellular processes required to establish a favorable maternal environment for pregnancy extend beyond the uterus, triggering a peripheral immune response. Altogether our findings underscore the role of the systemic immune environment in shaping the endometrial receptivity, an essential process for successful implantation and the establishment of pregnancy.

Additionally, our results indicate that upstream regulators play a crucial role in modulating the expression of these genes. Specially, the upstream regulators miR-16-5p (and other miRNAs with the seed sequence AGCAGCA) and miR-155-5p (miRNAs with the seed sequence UAAUGCU) were predicted to be inhibited in heifers classified as non-pregnant. Previous studies have reported the involvement of peripheral white blood cell-derived miRNAs in heifers that successfully conceived [69]. Furthermore, the upstream regulator AGT was predicted to be activated in pregnant heifers at PDC7, PCD14 and 1ETD7, which is consistent with previous research showing its activation in the endometrium on Days 6–8 post-estrus in cows confirmed pregnant after embryo transfer [7]. Moreover, studies have demonstrated that the growth factor AGT contributes positively to the establishment of endometrial receptivity and is predicted to be activated in pregnant cows [4,9]. While prior research focused on the endometrium, our results demonstrate that similar molecular changes can also be detected in WBCs, suggesting that systemic blood markers may reflect uterine receptivity throughout the estrous cycle.

This study highlighted the over expression of genes involved in potassium/ calcium signaling, metabolic pathways, tissue remodeling, and the positive regulation of angiogenesis in WBCs from animals in the pregnant group. These findings are consistent with those of Moran et al., who demonstrated that the cellular structure and local immunity of the endometrium play a crucial role in the fertility of dairy cows. The authors suggest that optimized management of inflammation and cytoskeletal processes could potentially improve conception rates [11]. Additionally, this work aligns with a previous study conducted on Angus-Simmental heifers, which were sampled at weaning and retrospectively classified as fertile or subfertile based on pregnancy diagnosis. This study identified a specific set of genes expressed in PBMCs related to potassium channels pathways, immune responses, hormonal regulation, and maturation of reproductive cells, which could predict the future fertility of heifers, even before they reach reproductive maturity [24]. These findings suggest that PBMCs gene expression profiles could serve as reliable biomarkers for assessing future reproductive potential, offering new

opportunities for early selection and enhanced management strategies in livestock fertility. Ultimately, the overall results of these studies reinforce the link between the systemic environment and the endometrium's capacity to sustain pregnancy. Taken together, our findings suggest that gene expression profiles in WBCs throughout the estrous cycle may reflect inflammatory, immune, and physiological states that could negatively affect embryo implantation and development in heifers. The interplay between inflammation and pregnancy success is well documented in humans, where finely regulated inflammation is essential for embryo implantation, but excessive or dysregulated immune cell inflammation can impair the successful establishment of pregnancy [70].

In summary, our findings support the concept of a tight dialogue between the endometrium and peripheral white blood cells, emphasizing the critical role of both systemic and local immune regulation in embryo implantation, development, and pregnancy establishment. Endometrial receptivity and the ability of a heifer to deliver a healthy offspring must be considered in an integrated manner. Our data provide new perspectives into monitoring female fertility, and underscore the need for further studies aimed at using whole peripheral blood or specific subtypes of circulating immune cells as sources of biomarkers to predict pregnancy outcomes in mammalian females.

## Conclusion

Pregnancy success involves a finely coordinated series of molecular and cellular events at both local and systemic levels. Our current study demonstrates that variations in the transcriptomic profiles of WBCs genes throughout the estrous cycle could serve as a potential predictive tool to evaluate the ability of a recipient female to support embryonic development to term following embryo transfer. Therefore, gene signatures of peripheral immune cells may provide a valuable practical approach for predicting embryo transfer success in recipient mammalian females. Additional large-scale studies are needed to validate and expand these results to ensure their broader applicability in reproductive management.

## Supporting information

**S1 Table. Holstein-Friesian Heifers involved in the experimental protocol. pregnancy phenotype and biochemical/ molecular analyses.**
(XLSX)

**S2 Table. DEGs in peripheral white blood cells isolated on Day 7 of preconception cycle (PCD7) from Holstein-Friesian heifers classified as pregnant (n=4) upon embryo transfer vs. non-pregnant (n=7) at conception cycles 1 and 2.**
(XLSX)

**S3 Table. DEGs in peripheral white blood cells isolated on Day 14 of preconception cycle (PCD14) from Holstein-Friesian heifers classified as pregnant (n=4) upon embryo transfer vs. non-pregnant (n=7) at conception cycles 1 and 2.**
(XLSX)

**S4 Table. DEGs in peripheral white blood cells isolated on Day 7 of the first conception cycle just before embryo transfer from Holstein-Friesian heifers classified as pregnant (n=4) upon embryo transfer vs. non-pregnant (n=7) at conception cycles 1 (1ETD7).**
(XLSX)

**S5 Table. Common DEGs between PCD7, PCD14 and 1ETD7 in peripheral white blood cells isolated from Holstein-Friesian heifers classified as pregnant (n=4) upon embryo transfer vs. non-pregnant (n=7) at conception cycles 1 and 2.**
(XLSX)

**S6 Table. IPA for networks associated with DEGs in peripheral white blood cells collected on PCD7 from Holstein-Friesian heifers classified as pregnant (n = 4) upon embryo transfer vs. non pregnant (n = 7) at conception cycles 1 and 2.**
(XLSX)

**S7 Table. IPA for networks associated with DEGs in peripheral white blood cells collected on PCD14 from Holstein-Friesian heifers classified as pregnant (n = 4) upon embryo transfer vs. non-pregnant (n = 7) at conception cycles 1 and 2.**
(XLSX)

**S8 Table. IPA for networks associated with DEGs in peripheral white blood cells collected on 1ETD7 from Holstein-Friesian heifers classified as pregnant (n = 4) upon embryo transfer vs. non-pregnant (n = 7) at conception cycles 1 and 2.**
(XLSX)

**S9 Table. Major functional categories and pathways (GO and KEGG) associated with genes significantly enriched in peripheral white blood cells from Holstein-Friesian heifers classified as non-pregnant upon embryo transfer vs. non-pregnant at conception cycles 1 and 2.**
(XLSX)

**S10 Table. Upstream regulators predicted to explain gene expression changes observed at PCD7 in peripheral white blood cells of Holstein-Friesian heifers classified as pregnant (n = 4) upon embryo transfer vs. non-pregnant (n = 7) at conception cycles 1 and 2.**
(XLSX)

**S11 Table. Upstream regulators predicted to explain gene expression changes observed at PCD14 in peripheral white blood cells from Holstein-Friesian heifers classified as pregnant (n = 4) upon embryo transfer vs. non-pregnant (n = 7) at conception cycles 1 and 2.**
(XLSX)

**S12 Table. Upstream regulators predicted to explain gene expression changes observed at 1ETD7 in peripheral white blood cells of heifers classified as pregnant (n = 4) upon embryo transfer vs. non-pregnant (n = 7) at conception cycles 1 and 2.**
(XLSX)

**S13 Table. Common upstream regulators at PCD7, PCD14 and 1ETD7 predicted to explain gene expression changes observed in peripheral white blood cells from Holstein-Friesian heifers classified as pregnant (n = 4) vs. non pregnant (n = 7).**
(XLSX)

## Acknowledgments

The authors thank -, Philippe Bolifraud for his technical support with sample processing, Prof. François Vialard for his helpful feedback on the manuscript, and Vincent Mauffré for valuable scientific discussions. We are also grateful to the Cellulaire Microarray Platform (IGBMC – Institute of Genetics, Molecular and Cellular Biology, Strasbourg, France; https://www.igbmc.fr/plateformes-technologiques/genomeast) for carrying out the microarray hybridizations, and to the @Bridge Genomic Analysis Platform (INRAE, Ile-de-France – Jouy-en-Josas – Antony Research Center, France) for providing access to the IPA software.

## Author contributions

**Conceptualization:** Mariam Raliou, Marie Margarete Meyerholz-Wohllebe, Kirsten Mense, Christophe Richard, David Smith, Peter Zieger, Hans-Joachim Schuberth, Marion Schmicke, Iain Martin Sheldon, Olivier Sandra.

**Data curation:** Mariam Raliou, Marie Margarete Meyerholz-Wohllebe, Doulaye Dembélé, Kirsten Mense, Maike Heppelmann, David Smith, Marion Schmicke.

**Formal analysis:** Mariam Raliou, Doulaye Dembélé.

**Funding acquisition:** Peter Zieger, Hans-Joachim Schuberth, Marion Schmicke, Iain Martin Sheldon, Olivier Sandra.

**Investigation:** Mariam Raliou, Iain Martin Sheldon.

**Methodology:** Mariam Raliou, Olivier Sandra.

**Project administration:** Isabelle Dieuzy-Labaye, Iain Martin Sheldon, Olivier Sandra.

**Resources:** Mariam Raliou.

**Supervision:** Mariam Raliou, Iain Martin Sheldon, Olivier Sandra.

**Validation:** Mariam Raliou, Iain Martin Sheldon, Olivier Sandra.

**Visualization:** Iain Martin Sheldon.

**Writing – original draft:** Mariam Raliou.

**Writing – review & editing:** Mariam Raliou, Marie Margarete Meyerholz-Wohllebe, Doulaye Dembélé, Kirsten Mense, Christophe Richard, Pascale Chavatte-Palmer, David Smith, Hans-Joachim Schuberth, Iain Martin Sheldon, Olivier Sandra.

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
