## [Decision Letter · Decision Letter 0]

4 Jun 2025

Dear Dr. Raliou,

Thank you for submitting your manuscript to PLOS ONE. After careful consideration, we feel that it has merit but does not fully meet PLOS ONE’s publication criteria as it currently stands. Therefore, we invite you to submit a revised version of the manuscript that addresses the points raised during the review process.

We look forward to receiving your revised manuscript.

Kind regards,

Muhammad Usman Mehmood

Academic Editor

PLOS ONE

Journal Requirements:

2. To comply with PLOS ONE submissions requirements, in your Methods section, please provide additional information regarding the experiments involving animals and ensure you have included details on methods of anesthesia and/or analgesia.

6. Please remove all personal information, ensure that the data shared are in accordance with participant consent, and re-upload a fully anonymized data set.

Additional Editor Comments :

Although the study is quite interesting, I believe it needs to be validated through a field trial.

Could you please explain the statistical analysis and clarify whether the data is normally distributed? Additionally, could you explain clearly what a repeated measures two-way ANOVA or a mixed-effects model entails?

Line 159, 2alpha should be underscript throughout the manuscript

Line 159, please write the exact dosage like mg/ml

Please write the complete kit details for hormones with a coefficient of variation.

Please provide complete details of the real-time quantitative PCR (qPCR) analysis.

Line 328, f-value is not the standard terminology, pls write it scientifically.

Please follow the journal pattren for refernces

Reviewers' comments:

Reviewer's Responses to Questions

**Comments to the Author**

1. Is the manuscript technically sound, and do the data support the conclusions?

Reviewer #1: Yes

Reviewer #2: Yes

2. Has the statistical analysis been performed appropriately and rigorously?

Reviewer #1: Yes

Reviewer #2: Yes

3. Have the authors made all data underlying the findings in their manuscript fully available?

Reviewer #1: Yes

Reviewer #2: Yes

4. Is the manuscript presented in an intelligible fashion and written in standard English?

Reviewer #1: Yes

Reviewer #2: Yes

Reviewer #1: Experiments were conducted rigorously, with appropriate controls and sample sizes.

The conclusions were appropriately presented, based on the data reported

Statistical analysis have been performed correctly and rigorously

The language in submitted articles is clear, correct, and unambiguous

The manuscript described multiple techniques and generated data that supports the conclusions.

Reviewer #2: The presented investigation is highly applied in nature and is executed. The developments regarding alternative tools for diagnosing pregnancy has the potential to significantly enhance the fertility goals in the cattle industry and could pave the way as a model for analogous applications within human medicine. The experimental design, in conjunction with the statistical analyses conducted, has been well designed and carried out in a very professional manner. However, it is my considered opinion that the discussion section would greatly benefit from the inclusion of more in-depth arguments, which would effectively introduce novel ideas or suggest new directions for further exploration. In my humble opinion, this article is worthy of acceptance, albeit with the recommendation for some minor improvements to be made specifically in the discussion section to improve its overall quality. I extend my sincere gratitude to the research team for their efforts in bringing forth innovative interventions that contribute meaningfully to our understanding of this important field.

**Do you want your identity to be public for this peer review?** For information about this choice, including consent withdrawal, please see our Privacy Policy

Reviewer #1: No

Reviewer #2: **Yes: ** Muhammad Shahzad (Xu Jia Song)

---

## [Author Response · Author response to Decision Letter 1]

18 Jul 2025

Please see the attached document titled "Responses to the Reviewers"

---

## [Decision Letter · Decision Letter 1]

6 Aug 2025

Gene profiles of peripheral white blood cells as potential predictors of pregnancy in embryo-recipient heifers

PONE-D-25-21659R1

Dear Dr. Mariam Raliou,

We’re pleased to inform you that your manuscript has been judged scientifically suitable for publication and will be formally accepted for publication once it meets all outstanding technical requirements.

Kind regards,

Muhammad Usman Mehmood

Academic Editor

PLOS ONE

Additional Editor Comments (optional):

I would be sincerely thankful to both reviwerers for their valuable reviews. Based on reviewers thoughtful feedback as well as my recommendations, I have made the decision to accept the manuscript for publication. Congratulate to all the authors.

Reviewers' comments:

Reviewer's Responses to Questions

**Comments to the Author**

Reviewer #2: All comments have been addressed

2. Is the manuscript technically sound, and do the data support the conclusions?

Reviewer #2: Yes

3. Has the statistical analysis been performed appropriately and rigorously?

Reviewer #2: Yes

4. Have the authors made all data underlying the findings in their manuscript fully available?

Reviewer #2: Yes

5. Is the manuscript presented in an intelligible fashion and written in standard English?

Reviewer #2: Yes

Reviewer #2: Dear Team,

This is a unique piece of work that will contribute something remarkable to the field of reproductive science. I would like you to further develop diagnostic kits based on this research.

**Do you want your identity to be public for this peer review?** For information about this choice, including consent withdrawal, please see our Privacy Policy

Reviewer #2: **Yes: ** Muhammad Shahzad (Xu JiaSong)

---

## [Editor Report · Acceptance letter]

PONE-D-25-21659R1

PLOS ONE

Dear Dr. Raliou,

I'm pleased to inform you that your manuscript has been deemed suitable for publication in PLOS ONE. Congratulations! Your manuscript is now being handed over to our production team.

Kind regards,

on behalf of

Dr. Muhammad Usman Mehmood

Academic Editor

PLOS ONE